# Language Problems and ADHD Symptoms: How Specific Are the Links?

**DOI:** 10.3390/brainsci6040050

**Published:** 2016-10-21

**Authors:** Erin Hawkins, Susan Gathercole, Duncan Astle, Joni Holmes

**Affiliations:** MRC Cognition and Brain Sciences Unit, Cambridge, CB2 7EF, UK; susan.gathercole@mrc-cbu.cam.ac.uk (S.G.); duncan.astle@mrc-cbu.cam.ac.uk (D.A.); joni.holmes@mrc-cbu.cam.ac.uk (J.H.)

**Keywords:** ADHD, attention, hyperactivity, learning, language, communication

## Abstract

Symptoms of inattention and hyperactivity frequently co-occur with language difficulties in both clinical and community samples. We explore the specificity and strength of these associations in a heterogeneous sample of 254 children aged 5 to 15 years identified by education and health professionals as having problems with attention, learning and/or memory. Parents/carers rated pragmatic and structural communication skills and behaviour, and children completed standardised assessments of reading, spelling, vocabulary, and phonological awareness. A single dimension of behavioural difficulties including both hyperactivity and inattention captured behaviour problems. This was strongly and negatively associated with pragmatic communication skills. There was less evidence for a relationship between behaviour and language structure: behaviour ratings were more weakly associated with the use of structural language in communication, and there were no links with direct measures of literacy. These behaviour problems and pragmatic communication difficulties co-occur in this sample, but impairments in the more formal use of language that impact on literacy and structural communication skills are tied less strongly to behavioural difficulties. One interpretation is that impairments in executive function give rise to both behavioural and social communication problems, and additional or alternative deficits in other cognitive abilities impact on the development of structural language skills.

## 1. Introduction

Symptoms of inattention and hyperactivity typically co-occur with poor communication skills [1,2] and low levels of literacy [3,4,5] in children with attention deficit hyperactivity disorder (ADHD). The same comorbidities are also found in the general school population [6,7] and in other neurodevelopmental conditions including autism spectrum disorder (ASD) [8], specific language impairment (SLI) [9], and dyslexia [10,11]. These overlapping symptom profiles may reflect dimensions of inattentive and hyperactive behaviour and cognitive difficulties that cut across traditional diagnostic categories [12,13,14]. In the current study we adopt a dimensional approach to test the specificity of the associations between these dimensions of behaviour, communication and literacy skills in a large sample of children receiving support from specialist services for difficulties in attention, learning and/or memory. The sample included a small number of children with diagnosed developmental disorders and a substantial number with sub-clinical difficulties. The atypical and heterogeneous nature of the sample enabled us to investigate the extent to which impairments in language and behaviour co-occurred in children with problems related to educational progress.

Difficulties in both the formal learning of language structure and the use of language in different contexts are common in developmental disorders [1,15,16,17]. Structural aspects of language include the use of phonology, semantics, syntax and morphology. These skills are important for literacy development [18,19] and for expressing and understanding spoken language in communication [20]. Pragmatic aspects of language involve the appropriate use of language in social communicative contexts such as maintaining appropriate topics, not talking excessively, turn-taking in conversations and interpreting non-verbal cues of others [21,22]. It is unclear whether impairments in pragmatic language arise as a secondary consequence of structural language problems [23] or instead reflect social or behavioural difficulties [7,24]. Evidence that children can have pragmatic language difficulties in the absence of problems with language structure supports the view that these two dimensions of language impairment may have different sources [22,25].

Profiles of structural and pragmatic language difficulties differ across specific diagnostic groups. For example, pragmatic language impairments are common in high-functioning autism [16] and ADHD [1,6,21,26], but structural language difficulties are more common in children with reading difficulties and SLI [12,14,27]. However, there is also considerable heterogeneity within each diagnostic category, as demonstrated for example by the prevalence of pragmatic language difficulties among children with SLI [28]. There is also substantial overlap between the linguistic profiles of children with different diagnoses, illustrated by the prevalence of pragmatic language difficulties in both children with ADHD and those with autism [1]. These two groups also show elevated symptoms of both inattention and hyperactivity typically common in both disorders [29,30,31], illustrating how symptom profiles of language and behaviour problems may not be specific to particular disorders.

Pragmatic communication problems are also associated with the severity of symptoms of inattention and hyperactivity in other clinical and typically developing populations [1,6,21]. A possible source of these co-morbid problems are the executive function difficulties frequently observed in children with attentional symptoms of ADHD [32,33]. It has been suggested that ADHD may arise from disruptions to two functionally distinct neurodevelopmental systems: cool cognitive-based functions that include working memory, planning, and inhibition, and which are associated with inattention; and hot affective processes that are associated with delay aversion and impulsive behaviours [34]. Effective pragmatic communication may also rely on both of these systems. Cool cognitive functions may be required to maintain information about the conversational topic and to produce coherent, well-planned and appropriate conversational speech [2,35,36], whilst affective processes may be required for appropriate turn taking, and to limit excessive talking [2,37]. Deficits in one or both of these dimensions may therefore impair pragmatic communication and cause problems with social and peer relationships [7,38]. Consistent with this, research has shown that children with elevated symptoms of inattention and hyperactivity have intact knowledge of pragmatics but problems in the executive skills needed to apply it in social contexts [2,6,39].

Impairments in language structure also co-occur with ADHD symptoms across different populations. Deficits in structural components of communication such as the use of syntax and phonology are present in children with ADHD [2]. Weak literacy skills are also associated with both elevated levels of inattention and hyperactivity in clinical and community samples [40,41,42,43,44], with stronger links with inattention [45,46,47,48]. A possible explanation for this association is that behavioural inattention disrupts the formal acquisition of reading skills through its impact on classroom behaviour, which reduces children’s ability to attend to direct instructions required for learning to read [49]. This model is supported by evidence that preschool inattention predicts later reading skills independently from other early markers of literacy development such as phoneme awareness and letter knowledge [40,41,50]. Difficulties paying attention may also have a direct effect on the development of structural language skills that are important for literacy development (e.g., phonological processing) [51,52,53]. An alternative possibility is that underlying deficits in executive functions such as working memory underlie both short attention spans and literacy problems [3,32,33].

Pragmatic and structural language problems encompassing poor structural communication and literacy therefore co-occur with the symptoms of ADHD, but evidence for the specificity and strengths of these relationships is mixed. To date, studies have focused on either diagnosed groups or community samples containing large numbers of typically developing children. The present study explored the relationship between these dimensions of impairment across diagnostic categories in a unique sample of struggling learners. The children had difficulties in attention, learning, or memory, and were receiving support from professionals working in children’s services, including speech and language therapists, educational psychologists, school-based special educational needs co-ordinators, mental health practitioners, and paediatricians. Approximately one third had received formal diagnoses (e.g., of ADHD, ASD, or dyslexia) and the remainder had sub-clinical difficulties of varying severity. Based on previous findings of an association between behavioural problems and pragmatic language difficulties we expected behaviour problems to be more highly linked with pragmatic than structural language skills overall, with additional specific associations between behavioural inattention and assessments of structural language skills in literacy.

Assessments included standardised tests of reading, spelling, phonological awareness and vocabulary, as well as parent ratings of children’s behaviour and pragmatic and structural communication skills. Because of the atypicality of this sample, a data driven approach was used to examine the relationships between behaviour and language skills. We sought to identify whether separate dimensions of inattentive and hyperactive behaviour problems were present in the sample, and the extent to which these behaviours were related to poor structural and pragmatic language skills.

## 2. Method

### 2.1. Participants

Participants were recruited from an ongoing study at the Cambridge Centre for Attention, Learning, and Memory (CALM) based in the MRC Cognition and Brain Sciences Unit in Cambridge, UK. Children were referred to the CALM research clinic for problems in one or more areas of attention, learning, and memory. The current sample consisted of the first 254 children seen in the clinic for whom there were complete data. The age range of the sample was 5:5–15:11 (M = 9:4, SD = 2:3), with 169 males and 85 females. No diagnosis was reported for the majority of the sample (*n* = 191). Twenty children had a diagnosis of ADHD, and a further 59 children had been identified with other developmental learning disorders (e.g., dyslexia, dyscalculia, autism spectrum disorders) or psychiatric diagnoses (see Table 1). Exclusion criteria for referrals to the clinic were: (i) significant and severe known problems in vision or hearing that are uncorrected; and (ii) a native language other than English. Ethical approval was provided by the Local NHS Research Ethics Committee, reference 13/EE/0157. Written informed consent was obtained from parents/guardians, and informed verbal assent was obtained from children.

### 2.2. Procedure

Children completed a battery of tests on a one-to-one basis with a researcher in a dedicated child-friendly testing room at the CALM clinic. These included a wide range of standardised assessments of learning and cognition. Regular breaks were included throughout the session. Testing was split over two sessions for children who struggled to complete the assessments in one sitting. Parents/carers completed a set of questionnaires relating to behaviour, learning, communication, and medical and family history in a family waiting area while their child was assessed. Measures of behaviour, communication skills, and literacy are reported here.

### 2.3. Measures

#### Behaviour

##### Parent ratings of behaviour

The Conners 3 Parent Short Form [54] is a questionnaire assessing behavioural and cognitive problems related to ADHD. Parents or caregivers rated 45 items as not true at all (0); just a little true (1); pretty much true (2); or very much true (4). Scores on these items formed six subscales consisting of Inattention, Hyperactivity, Learning Problems, Executive Function, Aggression, and Peer Relations. The sum of raw scores on each subscale was converted to a T-score (M = 50, SD = 10) for a normative sample 6:00–18:00 years of age. If children’s age fell outside the standardisation range the closest age match was used. T-scores ≥70 on an individual subscale indicate the child has very elevated behavioural problems for their age. T-scores 60–69 are within the elevated range, scores 40–59 are in the average range, and scores ≤39 indicate a child has fewer reported problems than average for their age.

### 2.4. Literacy

#### 2.4.1. Reading

The Single-Word Reading subtest on the Wechsler Individual Achievement Test II (WIAT-II) [55] assessed children’s reading abilities. Children read a list of words aloud that were scored by the examiner. Responses were coded as correct if they were pronounced correctly and fluently. Standard scores (M = 100, SD = 15) were derived from the raw scores, with a normative sample age range of 4:00–16:11.

#### 2.4.2. Spelling

The Spelling sub-test on the WIAT-II provided a measure of spelling attainment. Children were asked to spell words spoken one at a time by the examiner. Raw scores were converted to standard scores (M = 100, SD = 15) with a normative sample 4:00–16:11 years of age.

#### 2.4.3. Vocabulary

The Peabody Picture Vocabulary Test, Fourth Edition (PPVT-4) [56] assessed receptive vocabulary knowledge. Children were required to select one of four pictures showing the meaning of a spoken word. Standard scores were calculated (M = 100, SD = 15), and the normative sample had an age range of 2:6–90:00.

#### 2.4.4. Phonological awareness

The Alliteration subtest of the Phonological Assessment Battery (PhAB) [57] tested phonological awareness. Sets of three spoken words were read aloud and children were required to judge which two words started with the same sound. The number of correct responses formed raw scores, which were converted to standardised scores (M = 100, SD = 15). Children who were too young or unable to complete the Alliteration subtest were given the Alliteration Test with Pictures subtest. In this version children saw pictures corresponding to words named aloud by the researcher. Children had to indicate which two pictures corresponded to words starting with the same sound. Raw scores on the Alliteration Test with Pictures were converted to standardised scores (M = 100, SD = 15). If a child’s age fell outside the standardisation range (6:00–14:11) the closest age match was used to obtain a standardised score.

### 2.5. Communication

The Children’s Communication Checklist, second edition (CCC-2) [58] was used to measure communication skills. This 70-item questionnaire assessed language structure and form, and verbal and nonverbal pragmatic communication. Scaled scores (M = 10, SD = 3) were derived for 10 subscales that fell into three categories measuring different aspects of language use. The first four scales were (A) Speech; (B) Syntax; (C) Semantics; and (D) Coherence. These assessed language structure, vocabulary use, and discourse, and are areas of communication typically impaired in children with SLI. The next four scales were (E) Inappropriate Initiation; (F) Stereotyped Language; (G) Use of Context; and (H) Nonverbal Communication. These indexed verbal and nonverbal pragmatic communication skills. The final two scales, (I) Social relations; and (J) Interests, assessed aspects of language behaviour that are usually impaired in ASD. Clinically significant communication impairments on the CCC-2 are interpreted using percentiles rather than scaled scores. Scores below the 15th percentile, corresponding approximately to a scaled score of 6, are considered below normal limits (see [58] for further details). For the present purposes, scaled scores ≤6 are considered as corresponding to deficits. The normative sample had an age range of 4:0–16:11.

## 3. Results

### 3.1. Analysis Plan

To characterise our sample we first compared performance on each measure to the age-normed population mean, calculated the proportion of children in the deficit range on each measure, and checked for gender differences in performance. The following Bonferroni corrected significance thresholds were used to correct for multiple comparisons: behaviour, *p* < 0.008; communication, *p* < 0.005; literacy, *p* < 0.0125.

To assess the specificity and strength of links between ADHD behaviours and language skills we first conducted correlation analyses between each measure individual measure. Next, we used exploratory factor analysis (EFA) to reduce the dimensionality of the data and identify the underlying constructs of language and behaviour. One EFA was conducted for the behaviour measure. The language measures were entered into two separate EFAs, one for the parent/carer ratings of communication and another for the direct assessments of literacy. Principal axis factoring with direct Oblimin rotation was used to allow factors to be correlated. Factors were chosen based on an eigenvalue cut-off of 1 and examination of the scree plots.

Resulting factor scores were correlated to explore the associations between different dimensions of impairment. Partial correlations were also computed that controlled for age. The relative strength of the associations between each pair of factors was tested using Meng’s Z test, which is used to compare correlation coefficients [59]. For all analyses involving the communication data, scores were excluded for children whose CCC-2 ratings were identified as invalid due to disproportionately negative or positive responses [58].

### 3.2. Descriptive Statistics

Descriptive statistics are presented in Table 2. Relative to standardised age-norms, the sample had significantly poorer language and communication skills, and significantly elevated behavioural problems across all of the measures. Comparison of standard scores indicated that children had substantially more behaviour problems, poorer spoken communication and poorer reading, spelling, and phonological skills. This is to be expected given how the sample was recruited. Independent *t*-tests comparing scores for males and females on each measure indicated gender differences only on the literacy measures of vocabulary and phonological skills, in which males scored higher. No gender differences were present across the behaviour and communication measures, or in the literacy measures of reading and spelling.

### 3.3. Correlational Analyses

Correlations between the behaviour and communication measures are presented in Table 3. The majority of the behaviour measures showed a significant negative correlation with the communication measures, indicating that children with poorer behaviour had weaker communication skills. Correlations between Inattention and Speech, and between Executive Function, Speech and Syntax, were not significant.

Table 4 shows the correlations between behaviour and the literacy measures. Literacy skills were negatively associated with ratings of Learning Problems, showing that children rated as poorer learners had lower scores on the direct measures of literacy skills. There were no other significant relationships between behavioural difficulties and literacy.

The correlations between the literacy and communication measures are presented in Table 5. Reading, spelling, and vocabulary were positively correlated with the majority of the communication measures, indicating that children rated as having better communication skills performed better on these direct assessments of literacy. Phonological awareness was positively correlated only with the parent-rated structural communication scales of speech, syntax, and semantics.

### 3.4. Exploratory Factor Analyses

Table 6, Table 7 and Table 8 show the component matrices for parent ratings of behaviour and communication, and the literacy measures respectively. In all factor analyses, eigenvalues below 0.40 were suppressed.

A one-factor solution emerged for the behaviour ratings, which explained 37.94% of the variance. The Inattention and Hyperactivity subscales loaded most highly on this factor, with moderate loadings of Executive Function, Aggression and Peer Relations. The Learning Problems subscale was excluded as it did not have a sufficiently high loading.

A two-factor solution emerged for the communication scale. The six subscales measuring pragmatic and social communication loaded on Factor 1, which accounted for 59.71% of the variance. Three of the four subscales tapping structural communication skills loaded on Factor 2, explaining 9.03% of the variance. Factor 1 was therefore linked with pragmatic and social aspects of communication, and Factor 2 with structural communication skills. The Coherence variable loaded on both factors (Factor 1= 0.553; Factor 2 = 0.423), reflecting its sensitivity to both pragmatic and structural communication problems [16,58].

For the direct measures of literacy, a one-factor solution emerged that explained 48.56% of the variance. Reading and Spelling loaded most highly on this factor, with moderate loadings from Vocabulary and Phonological Alliteration.

### 3.5. Correlations between Factor Scores

The behaviour factor was strongly negatively associated with both the pragmatic (*r* = −0.617, *p* < 0.001) and structural (*r* = −0.322, *p* < 0.001) communication factors, indicating that children with more behavioural problems had poorer communication skills (see Table 9 and Figure 1). The relationship between behaviour problems and pragmatic communication skills was significantly stronger than the association between behaviour and structural communication skills (Meng’s Z = −6.255, *p* < 0.001). In line with this, there was no relationship between the behaviour factor and direct measures of language structure as captured by the literacy factor (*r* = −0.097, *p* > 0.14).

Literacy was significantly related to both structural (*r* = 0.455, *p* < 0.001) and pragmatic (*r* = 0.152, *p* < 0.05) communication factors, indicating that children with better performance on the direct measures of structural language had better parent-reported communication skills. The association between literacy and structural communication skills was significantly stronger than the association between literacy and pragmatics (Meng’s Z = −5.747, *p* < 0.001), indicating that the direct assessments of literacy shared more variance with the parent ratings of structural language in communication (see Figure 2). These patterns of association were unaffected by controlling for age.

## 4. Discussion

This study explored associations between the behavioural symptoms of ADHD and pragmatic and structural language abilities in a heterogeneous sample of children receiving support for problems in attention, learning, or memory. A bottom-up data driven approach identified dimensions of behaviour and language characterising the sample. This yielded a single dimension of behavioural problems that included inattention, hyperactivity and other behaviour problems commonly associated with ADHD. Although symptoms of inattention and hyperactivity can be diagnostically separate [60], it was not possible to differentiate them in the current sample. Three related dimensions of language emerged. One captured pragmatic communication skills and two encompassed children’s use of language structure. These were structural communication and literacy. Behaviour problems were strongly associated with pragmatic aspects of language, but less so with structural communication skills and not at all with literacy.

These differences in the relationship between the pragmatic and structural aspects of language function and behaviour are consistent with the idea that language impairments may have different sources [22,25]. Co-occurring behaviour difficulties and pragmatic communication problems may be underpinned by a common deficit in executive functioning. Pragmatic skills such as taking turns in conversation and not talking excessively, and the ability to control hyperactive/impulsive behaviours may both rely on hot executive skills like self-regulation and inhibition [2,37]. Likewise, features of social interaction such as monitoring and maintaining appropriate topics of conversation, and planning coherent speech, may draw on the same cool executive functions needed to maintain focussed attention (e.g., working memory) [61]. Consistent with this interpretation, executive function problems were present in the behaviour ratings in our sample. An alternative explanation is that there is a direct link between poor behaviour and weak pragmatic skills. By this account, difficult behaviour might directly limit opportunities for social interaction at home or school, impairing the development of social language skills [24,38]. Our sample had problems with peer relationships, signalling that they may indeed have had fewer opportunities to practice or develop pragmatic language skills.

Impairments in language structure may arise from different or additional sources to pragmatic and behavioural problems. Consistent with this, there was a weaker association between structural communication and behaviour than between pragmatics and behaviour. Moreover, literacy was unrelated to behaviour. This was unexpected given previous evidence linking ADHD symptoms, and in particular inattentive behaviour, with language and reading abilities [41,49]. One possibility is that the cognitive deficits limiting the acquisition and development of linguistic knowledge and skills are different to the executive function problems that are affecting behavioural control and social communication in our sample. These additional or alternative deficits might include problems with phonological processing. Supporting this idea, in the current study children’s phonological awareness and abilities in reading and spelling were strongly associated, but there was no link between their behavioural attention problems and phonological skills. Phonological processing skills are important for literacy development [14] and difficulties with phonological processing influence the acquisition of letter-sound correspondences that are important for learning to read [27,40]. Phonological processing impairments are commonly observed in children with SLI who have structural language impairments [12]. However, children with elevated symptoms of ADHD or pragmatic language difficulties tend to have intact phonological processing skills unless they have co-morbid problems with language structure [62,63], suggesting phonological deficits may be more closely associated with broader structural language problems than pragmatic difficulties.

There were stronger links between the parent-reported measures of language and behaviour than between parent-rated behaviour and the direct tests of literacy. This raises the possibility that the associations between parent-rated communication and behaviour simply reflect common variance in subjective reports from parents/carers of children who are receiving support from professional services. Although we cannot rule out this possibility, several features of the data suggest the parent ratings provided meaningful measurements of children’s behaviour and language skills. Firstly, the parent ratings of communicative language were related to the standardised literacy assessments, with a stronger relationship between parent-rated structural communication and direct literacy measures than between pragmatic communication and literacy. Second, these parent-rated communication scores similarly showed different strengths of association with parents’ behaviour ratings. Finally, parents’ views of their children’s learning abilities, as measured by a subscale of the behaviour checklist, correlated with the direct measures of literacy despite there being no other relationships between these measures and other scales on the behaviour checklist. Taken together, the specificity of these links speaks against a common variance in the parent measures underpinning the results.

## 5. Conclusions

In summary, the phenotype of these children with problems in attention, learning or memory was characterised by a broad dimension of inattentive and hyperactive behaviour that was strongly associated with pragmatic language difficulties, and more weakly associated with difficulties in the use of language structure. The comorbidities between behavioural symptoms and pragmatic language problems may reflect a cluster of developmental problems underpinned by deficits in executive function difficulties [2], while problems with language structure may arise through difficulties in phonological processing. From a practical perspective, these findings suggest that different profiles of language impairment may require different intervention approaches. Pragmatic communication problems may benefit from behavioural interventions, such as psychosocial therapy or executive function-based interventions [64,65]. Approaches targeting other cognitive difficulties, such as phonological weaknesses [66] might be more beneficial than behavioural interventions for problems in language structure.

## Figures and Tables

**Figure 1 brainsci-06-00050-f001:**
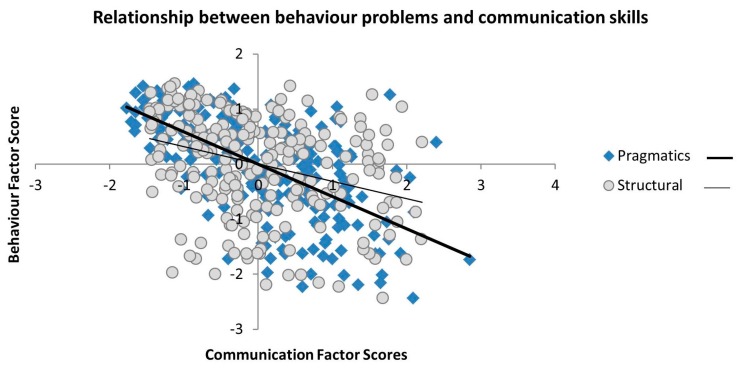
Scatterplots of the correlations between the pragmatic and structural communication factors and behaviour problems.

**Figure 2 brainsci-06-00050-f002:**
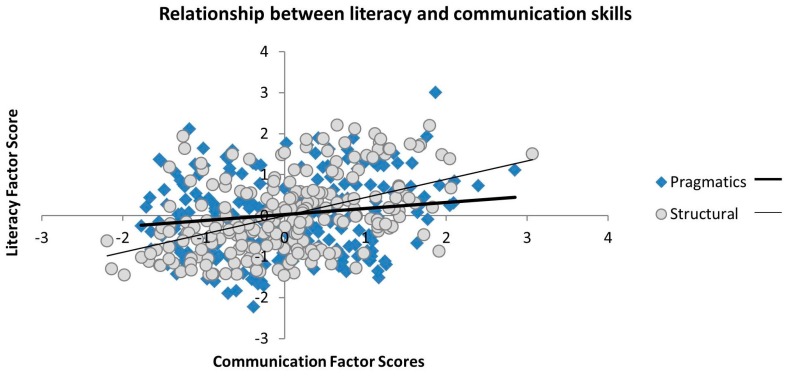
Scatterplots of the correlations between the pragmatic and structural communication factors and literacy.

**Table 1 brainsci-06-00050-t001:** Sample referral and diagnoses.

**Referrer**	**Male**	**Female**	**Total**
SENCo	103	58	161
Specialist Teacher	8	5	13
Educational Psychologist	4	2	6
Speech and Language Therapist	13	9	22
Clinical Psychologist	13	7	20
Paediatrician	22	4	26
**Diagnosis**	**Male**	**Female**	**Total**
None	105	59	164
ADD	2	2	4
ADHD	16	4	20
ASD	11	1	12
DAMP	2	1	3
Depression	0	2	2
Dysgraphia	1	0	1
Dyslexia	13	5	18
Dyspraxia	6	3	9
FASD	1	2	3
Generalised Developmental Delay	1	1	2
Global Delay, Dyspraxia	1	0	1
Hyperactivity	1	0	1
OCD	0	1	1
Social Anxiety Disorder, Depression	0	1	1
Tourettes	1	0	1
**Primary Reason for Referral**	**Male**	**Female**	**Total**
Attention	34	12	46
Literacy	30	11	41
Maths	6	3	9
Language	16	7	23
Poor academic progress	53	41	94
Memory	20	8	28
Anxiety	1	0	1

Note. The following datapoints were missing: Referral route, 6 data points; Diagnosis, 11 data points; Primary Reason for Referral, 12 datapoints. Abbreviations: SENCo–Special Educational Needs Coordinator; ADD–Attention Deficit Disorder; ADHD–Attention Deficit Hyperactivity Disorder; ASD–Autism Spectrum Disorder; DAMP–Deficits in Attention and Motor Perception; FASD–Fetal Alcohol Spectrum Disoder; OCD–Obsessive Compulsive Disorder.

**Table 2 brainsci-06-00050-t002:** Sample descriptive statistics.

	Whole Sample							Gender Split						
	*N*	Mean	SD	Effect Size †	Percentage of Children in Deficit Range	Skewness	Kurtosis	Male Mean	SD	Female Mean	SD	t	p	d
Age in months	254	111.830	26.804					111.69	26.746	112.12	27.075	−0.119	0.905	−0.016
Behaviour (Conners)														
Inattention	253	78.522	12.281	2.560	88.9	−0.978	−0.014	79.435	11.468	76.718	13.640	1.576	0.117	0.216
Hyperactivity	254	70.413	16.098	1.564	71.3	−0.293	−1.230	71.864	15.934	67.529	16.127	2.037	0.043	0.270
Learning Problems	254	76.228	12.017	2.383	89.8	−0.540	−0.789	74.929	12.132	78.812	11.420	−2.454	0.015	−0.330
Executive Function	254	72.047	13.235	1.898	81.5	−0.412	−0.767	72.308	12.412	71.529	14.801	0.417	0.678	0.057
Aggression	254	60.961	16.778	0.819	43.7	0.668	−1.036	61.615	16.773	59.659	16.811	0.877	0.382	0.117
Peer Relations	254	70.421	18.708	1.423	63.4	−0.377	−0.979	71.811	18.639	67.659	18.646	1.675	0.095	0.223
Communication (CCC-2)														
Speech	248	5.528	3.991	−1.279	61.7	0.300	−1.056	5.484	3.948	5.863	4.217	−0.663	0.508	−0.093
Syntax	248	5.436	4.131	−1.280	63.9	0.340	−1.083	5.369	4.021	5.849	4.462	−0.813	0.417	−0.113
Semantics	248	4.492	3.421	−1.716	76.5	0.873	0.302	4.331	3.350	4.932	3.713	−1.222	0.223	−0.170
Coherence	248	4.682	3.017	−1.768	75.7	0.741	0.506	4.618	3.083	4.822	3.133	−0.465	0.642	−0.066
Inappropriate initiation	248	6.077	3.053	−1.296	57.4	0.509	0.057	5.834	3.065	6.507	3.056	−1.550	0.122	−0.220
Stereotyped language	246	6.155	3.401	−1.202	53.9	0.300	−0.528	6.089	3.483	6.380	3.437	−0.587	0.558	−0.084
Use of context	248	4.617	3.630	−1.624	72.6	0.733	0.076	4.612	3.770	4.822	3.645	−0.398	0.691	−0.057
Nonverbal communication	248	5.371	3.520	−1.420	64.3	0.499	−0.392	5.312	3.614	5.534	3.363	−0.443	0.658	−0.064
Social relations	248	5.488	4.128	−1.266	58.7	0.278	−1.141	5.229	4.255	6.356	4.087	−1.893	0.060	−0.270
Interests	248	6.137	2.939	−1.301	61.7	0.828	0.866	5.847	2.820	6.890	3.247	−2.487	0.014	−0.344
Literacy														
Reading	253	85.455	17.354	−0.899	50.2	0.103	−0.449	86.148	17.989	84.060	16.012	0.901	0.368	0.123
Spelling	251	82.406	15.111	−1.169	62.2	-0.293	3.217	82.249	15.911	82.732	13.400	−0.237	0.813	−0.033
Vocabulary	250	97.172	15.631	−0.185	22.0	−0.315	1.330	99.898	14.949	91.687	15.618	4.030	0.000 **	0.537
Alliteration	251	90.697	10.044	−0.743	32.3	−0.616	−0.823	92.132	9.393	87.845	10.724	3.112	0.002 **	0.426

Note. † The effect size of the difference between the normative sample mean and the current sample mean. The deficit range on each measure was scores falling more than 1 SD below the normative sample mean. ** denotes significance at the 0.01 level, two-tailed.

**Table 3 brainsci-06-00050-t003:** Correlations between behaviour ratings and communication subscales.

Measure	1	2	3	4	5	6	7	8	9	10	11	12	13	14	15
1. Inattention															
2. Hyperactivity/impulsivity	0.573 **														
3. Learning problems	0.324 **	0.172 **													
4. Executive function	0.596 **	0.419 **	0.330 **												
5. Aggression	0.338 **	0.530 **	0.197 **	0.282 **											
6. Peer Relations	0.369 **	0.332 **	0.220 **	0.342 **	0.372 **										
7. Speech	−0.114	−0.141 *	−0.315 **	−0.127	−0.267 **	−0.177 **									
8. Syntax	−0.137 *	−0.180 **	−0.361 **	−0.106	−0.295 **	−0.201 **	0.689 **								
9. Semantic	−0.198 **	−0.183 **	−0.398 **	−0.139 *	−0.207 **	−0.291 **	0.574 **	0.617 **							
10. Coherence	−0.316 **	−0.299 **	−0.338 **	−0.300 **	−0.350 **	−0.405 **	0.543 **	0.635 **	0.690 **						
11. Inappropriate initiation	−0.429 **	−0.539 **	−0.302 **	−0.376 **	−0.448 **	−0.474 **	0.351 **	0.417 **	0.523 **	0.702 **					
12. Stereotyped language	−0.382 **	−0.384 **	−0.346 **	−0.344 **	−0.402 **	−0.411 **	0.444 **	0.523 **	0.597 **	0.708 **	0.704 **				
13. Use of context	−0.362 **	−0.319 **	−0.433 **	−0.283 **	−0.432**	−0.536 **	0.536 **	0.612 **	0.637 **	0.768 **	0.696 **	0.734 **			
14. Nonverbal communication	−0.332 **	−0.400 **	−0.256 **	−0.349 **	−0.462**	−0.528 **	0.454 **	0.493 **	0.478 **	0.663 **	0.647 **	0.677 **	0.750 **		
15. Social	−0.402 **	−0.407 **	−0.260 **	−0.404 **	−0.541 **	−0.676 **	0.335 **	0.389 **	0.459 **	0.623 **	0.625 **	0.620 **	0.708 **	0.755 **	
16. Interests	−0.328 **	−0.378 **	−0.202 **	−0.305 **	−0.350 **	−0.455 **	0.250 **	0.297 **	0.444 **	0.584 **	0.716 **	0.605 **	0.610 **	0.601 **	0.650 **

Note. 1–6 are the Conners behaviour scales and 7–16 are the CCC-2 communication scales. The correlations including the CCC-2 measures exclude participants for whom the CCC-2 ratings were identified as invalid due to disproportionately positive or negative responses. *N* = 254 for correlations within the behaviour subscales, except for correlations with inattention (*N* = 253). *N* = 230 for all correlations with the communication subscales, except for inattention with communication (*N* = 229). All correlations with stereotyped language had *N* = 228, with the exception of the correlation with inattention which had *N* = 227. * denotes significance at the 0.05 level; ** denotes significance at the 0.01 level, two-tailed.

**Table 4 brainsci-06-00050-t004:** Correlations between behaviour ratings and literacy measures.

Measure	1	2	3	4	5	6	7	8	9
1. Inattention									
2. Hyperactivity/impulsivity	0.573 **								
3. Learning problems	0.324 **	0.172 **							
4. Executive function	0.596 **	0.419 **	0.330 **						
5. Aggression	0.338 **	0.530 **	0.197 **	0.282 **					
6. Peer Relations	0.369 **	0.332 **	0.220 **	0.342 **	0.372 **				
7. Reading	0.008	−0.022	−0.554 **	−0.041	−0.094	−0.002			
8. Spelling	−0.050	−0.038	−0.543 **	−0.103	−0.094	−0.114	0.741 **		
9. Vocabulary	−0.032	0.009	−0.343 **	0.088	−0.123	−0.074	0.450 **	0.325 **	
10. Phonological Alliteration	−0.010	0.027	−0.311 **	−0.039	−0.032	0.005	0.437 **	0.301 **	0.437 **

Note. 1–6 are the Conners behaviour scales and 7–10 are the literacy measures. The sample size for the behaviour scales is as in Table 4. The sample sizes for the literacy measures were as follows. Vocabulary *N* = 250; spelling *N* = 251, except for with vocabulary (*N* = 247); reading *N* = 253, except for with vocabulary (*N* = 249) and spelling (*N* = 250); alliteration *N* = 251 except for with vocabulary (*N* = 247), spelling (*N* = 249), and reading (*N* = 250). * denotes significance at the 0.05 level; ** denotes significance at the 0.01 level, two-tailed.

**Table 5 brainsci-06-00050-t005:** Correlations between literacy measures and communication subscales.

Measure	1	2	3	4	5	6	7	8	9	10	11	12	13
1. Reading													
2. Spelling	0.743 **												
3. Vocabulary	0.461 **	0.320 **											
4. Phonological Alliteration	0.429 **	0.274 **	0.426 **										
5. Speech	0.353 **	0.336 **	0.335 **	0.233 **									
6. Syntax	0.414 **	0.358 **	0.438 **	0.311 **	0.689 **								
7. Semantic	0.368 **	0.302 **	0.363 **	0.154 *	0.574 **	0.617 **							
8. Coherence	0.236 **	0.225 **	0.288 **	0.105	0.543 **	0.635 **	0.690 **						
9. Inappropriate initiation	0.136 *	0.145 *	0.193 **	0.029	0.351 **	0.417 **	0.523 **	0.702 **					
10. Stereotyped language	0.173 **	0.203 **	0.265 **	0.044	0.444 **	0.523 **	0.597 **	0.708 **	0.704 **				
11. Use of context	0.287 **	0.297 **	0.362 **	0.125	0.536 **	0.612 **	0.637 **	0.768 **	0.696 **	0.734 **			
12. Nonverbal communication	0.073	0.112	0.157 *	−0.007	0.454 **	0.493 **	0.478 **	0.663 **	0.647 **	0.677 **	0.750 **		
13. Social	0.076	0.123	0.162 *	−0.073	0.335 **	0.389 **	0.459 **	0.623 **	0.625 **	0.620 **	0.708 **	0.755 **	
14. Interests	0.077	0.093	0.082	−0.152 *	0.250 **	0.297 **	0.444 **	0.584 **	0.716 **	0.605 **	0.610 **	0.601 **	0.650 **

Note. The correlations including the CCC-2 measures exclude participants for whom the CCC-2 ratings were identified as invalid due to disproportionately positive or negative responses. The correlations with literacy here thus only include participants with valid CCC-2 ratings. * denotes significance at the 0.05 level; ** denotes significance at the 0.01 level, two-tailed.

**Table 6 brainsci-06-00050-t006:** Exploratory factor analysis on the Conners behaviour subscales.

Behaviour Measures	Factor 1
Inattention	0.783
Hyperactivity/impulsivity	0.707
Executive Functions	0.664
Aggression	0.554
Peer Relations	0.517
Learning Problems	

Note. Factor loadings lower than 0.4 are not shown.

**Table 7 brainsci-06-00050-t007:** Exploratory factor analysis on the CCC-2 communication subscales.

Communication Measures	Factor 1	Factor 2
Interests	0.892	
Social	0.838	
Inappropriate Initiation	0.835	
Nonverbal	0.749	
Stereotyped language	0.682	
Use of context	0.659	
Coherence	0.553	0.423
Syntax		0.856
Speech		0.814
Semantics		0.579

Note. Factor loadings lower than 0.4 are not shown.

**Table 8 brainsci-06-00050-t008:** Exploratory factor analysis on the literacy measures.

Literacy Measures	Factor 1
Reading	0.950
Spelling	0.712
Vocabulary	0.528
Alliteration	0.505

Note. Factor loadings lower than 0.4 are not shown.

**Table 9 brainsci-06-00050-t009:** Correlations between factor scores.

Factor	1	2	3
1. Behaviour Factor			
2. Literacy Factor	−0.091		
3. Pragmatic and Social Communication Factor	−0.617 **	0.168 *	
4. Structural Communication Factor	−0.322 **	0.456 **	0.656 **

Note. These correlations were unchanged when controlling for children’s age in months. * denotes significance at the 0.05 level; ** denotes significance at the 0.01 level, two tailed.

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
