# Peer review of "Language Problems and ADHD Symptoms: How Specific Are the Links?"

_brainsci, 2016, doi:10.3390/brainsci6040050_

Round 1

Reviewer 1 Report

This is a well written paper examining the relationship between ADHD symptoms and aspects of language functioning (N=254). The study has a number of strengths including the consideration of both inattention and hyperactivity/impulsivity, as well as consideration of both structural and pragmatic forms of language skills. Limitations include the brief measurement of language skills, the broad age range of participants (5-15 years), the absence of a control group and the collection of only parent-reported behavioural measures.

My main questions about the paper relate to the analytic methods used:

-       It would be helpful if the analysis plan section could be structured around the aims of the study.

-       Can the authors justify the use of exploratory factor analysis as opposed to confirmatory factor analysis, especially given the known factor structure of the Conners and CCC-2 measures?

-       T-tests and correlational analyses are the main analyses used. There is no consideration of potential confounding variables in the analyses i.e. are there other factors that could be accounting for the relationships observed between behaviour and language/communication variables e.g., other difficulties like learning disorders, ASD etc. Furthermore, do these relationships hold when taking into account other variables including child age, gender and socio-economic status?

-       The results section splits results into communication and language attainment but a theoretical justification for this isn’t provided in the introduction section for this split. Rather the introduction focuses on splitting communication difficulties into pragmatic and structural difficulties.

Some additional comments:

-       The authors should be explicit in their methods when describing measures about whether all measures were validated & have normative data available for children aged 5 to 15 years. Currently this is only specifically noted for one measure.  

-       The lack of relationship between inattention and literacy attainment is highly unusual. The authors provide a plausible explanation for this in their results section but this really is at odds with so many previous findings.

-       The rationale for comparing the ADHD versus High Hyperactivity groups was unclear. This approach also seems at odds with the final sentence of paragraph one in the introduction about the adoption of a dimensional approach. This comparison raises many questions given that no information is provided in the manuscript about how ADHD diagnoses were made and the validity of these diagnoses. If these analyses are retained, more information is needed in the methods about the validity of the diagnosis. Alternatively this section of the paper could be omitted without impacting on the main messages of the paper.

-       No demographic characteristic are provided about the sample. The absence of these data makes it difficult to understand the generalisability of the study findings.

Thank you for the opportunity to review this interesting work.

Author Response

Thank you for your valuable comments on our submission. They have motivated a complete re-think of the paper and re-analysis of the data. We have made three major changes to the manuscript as a result of comments from both Reviewers:

i) In response to concerns raised by both reviewers about how we defined language in our original submission, we have developed the theory in the introduction to make a clear distinction between structural and pragmatic aspects of language function. In doing this we have reframed the manuscript and made it explicit that structural language skills are assessed through parent-ratings of structural communication skills and standardised measures of literacy, while pragmatics are assessed through parent-rated abilities.

ii) Reviewer 1 made an important point that the subgroup comparison did not add to the main message of the paper. The subgroup analyses have been removed to streamline the manuscript.

iii) Both reviewers wanted us to justify our motivation for using exploratory factor analyses and to provide further information about how the different factor solutions were selected for each set of measures. In our previous analysis of the behaviour measures we forced a two-factor solution to separate the two symptom dimensions of ADHD (hyperactivity and inattention). These two factors were very highly correlated (r = .676). For the direct measures of language we forced a two-factor solution to tease apart two distinct features of language attainment (phonological abilities and literacy), and for the parent-ratings of communication we allowed factors to emerge from the data as they were consistent with a distinction between pragmatic and structural communication skills. In the revised analysis we have used an entirely bottom-up approach to allow factor solutions to emerge from the data. This has produced a much simplified pattern of results. A single dimension of behaviour problems was identified. This was strongly associated with a language factor linked with pragmatics, but less so with a structural communication factor and not at all with a factor capturing literacy abilities. We hope the resulting message of the paper is clear: behavioural problems are more strongly related to pragmatic language difficulties than to structural language difficulties.

Due to the substantial revisions we have not included tracked changes on the manuscript. Detailed responses are provided below.

1. This is a well written paper examining the relationship between ADHD symptoms and aspects of language functioning (N=254). The study has a number of strengths including the consideration of both inattention and hyperactivity/impulsivity, as well as consideration of both structural and pragmatic forms of language skills. Limitations include the brief measurement of language skills, the broad age range of participants (5-15 years), the absence of a control group and the collection of only parent-reported behavioural measures.

                Given the small number of language measures included in the study, the tests of reading, spelling, vocabulary, and phonological alliteration are now referred to as measures of literacy. They are used as a proxy for children’s knowledge of language structure. These tasks loaded on a single factor, which was highly correlated with a factor tapping the use of language structure in communication.  

Partial correlations controlling for age were conducted and are reported in the Results section on page 10. These did not alter the patterns of association between the dimensions of behaviour and language.

                A control group was not included because we were not interested in comparing performance between our atypical sample and a typically developing group. Instead, we used a dimensional approach to investigate relationships between behaviour and language across a heterogeneous sample of children with learning difficulties. Substantial revisions have been made to the introduction to make this clearer, and full diagnostic information about the sample is provided in Table 1.
                Our assessments were limited to those administered in the CALM clinic protocol for which we have ethical approval. This included only parent-rated measures of behaviour. The main concern arising from this is whether the association between parent-ratings of behaviour and communication reflect meaningful relationships or common method variance (a point raised by Reviewer 2). Several features of the data suggest they provide meaningful measurements. These are described fully on page 12 in paragraph 2, but one pertinent point is that parent ratings of learning abilities, as measured by a subscale of the behaviour checklist, correlated with the direct measures of literacy. Finding specific links between subjective and objective measures of learning provides some validity for the parent-ratings of behaviour.   

2. It would be helpful if the analysis plan section could be structured around the aims of the study.

                The analysis plan has been re-structured around the aims of the study, which were i) to characterise the sample relative to standardised age-norms, ii) to explore links between the individual measures of language and behaviour, iii) to identify dimensions of language and behaviour through exploratory factor analyses and iv) to investigate links between these dimensions using correlational analyses (see Analysis Plan, page 8).

3. Can the authors justify the use of exploratory factor analysis as opposed to confirmatory factor analysis, especially given the known factor structure of the Conners and CCC-2 measures?

                Due to the atypical and heterogeneous nature of the sample it was important to use exploratory factor analyses as we could not assume the distributional properties of the data would fit the known factor structures of the Conners and CCC-2. The key aim of the revised paper is to use an entirely bottom-up data-driven approach, which is commensurate with the use of exploratory rather than confirmatory methods. 

4. T-tests and correlational analyses are the main analyses used. There is no consideration of potential confounding variables in the analyses i.e. are there other factors that could be accounting for the relationships observed between behaviour and language/communication variables e.g., other difficulties like learning disorders, ASD etc. Furthermore, do these relationships hold when taking into account other variables including child age, gender and socio-economic status?

                The potentially confounding effects of age and gender have been investigated. Controlling for age did not affect the relationships between language and behaviour (see page 10, line 286). Gender differences are reported in Table 2 and on page 8, lines 232-235. Males scored significantly higher than females on measures of vocabulary and phonological alliteration, but there were no other significant effects of gender for the individual tasks or in the factor scores (page 10) rendering it unlikely that the relationships between language and behaviour were driven by gender effects.

                It was not possible to investigate differences related to diagnosis due to the small number of children in each diagnostic group. Furthermore, a high proportion of the sample had sub-clinical difficulties, meaning any comparisons across diagnostic groups would not be meaningful. SES data was not available.

5. The results section splits results into communication and language attainment but a theoretical justification for this isn’t provided in the introduction section for this split. Rather the introduction focuses on splitting communication difficulties into pragmatic and structural difficulties.

                In line with much of the research into language abilities in children with developmental disorders, the language measures have been reconceptualised as tapping pragmatic and structural skills (see page 3, lines 58-79 for theoretical justification). The use of structural language was measured by both the literacy assessments and parent-rated structural communication skills. Pragmatics were assessed by parent-ratings.

Some additional comments:

6. The authors should be explicit in their methods when describing measures about whether all measures were validated & have normative data available for children aged 5 to 15 years. Currently this is only specifically noted for one measure. 

                Normative data was available for all measures. The age range of the normative sample is now reported for each measure (under Measures in the Method section).

7. The lack of relationship between inattention and literacy attainment is highly unusual. The authors provide a plausible explanation for this in their results section but this really is at odds with so many previous findings.

                This was an unexpected finding, which is at odds with associations commonly reported between attention and literacy in typically developing samples or specific diagnostic groups. This is the first large-scale investigation into the relationships between these two dimensions of difficulty in a highly atypical and transdiagnostic group, which may explain why the patterns are different to previous findings.  It is suggested that the absence of this association in the current sample might reflect the fact that behaviour and literacy problems arise through different sources in our sample, or that weak literacy skills are underpinned by additional cognitive impairments that do not cause behavioural difficulties (e.g. phonological processing problems).  This point is discussed on pages 11-12, lines 318-336.

8. The rationale for comparing the ADHD versus High Hyperactivity groups was unclear. This approach also seems at odds with the final sentence of paragraph one in the introduction about the adoption of a dimensional approach. This comparison raises many questions given that no information is provided in the manuscript about how ADHD diagnoses were made and the validity of these diagnoses. If these analyses are retained, more information is needed in the methods about the validity of the diagnosis. Alternatively this section of the paper could be omitted without impacting on the main messages of the paper.

                The subgroup analyses have been removed from the paper.

9. No demographic characteristic are provided about the sample. The absence of these data makes it difficult to understand the generalisability of the study findings.

                The available demographic data (age, gender, diagnosis) are provided in Table 1 to help characterise the sample.

Reviewer 2 Report

Rationale

The authors provide a reasonable justification for examining the associations between language and ADHD symptoms. 

Method

The measures used in this study are common in the ADHD literature.  The measures reflect a mixture of direct testing for the language measures and parent report for behavior and communication.  It is very likely that there is a sizable amount of covariance across the parent measures (common method variance) that reflect the parent’s response set and this is not present in the direct measures of language.  Throughout the paper, the authors keep the directly tested language measures separate from the measures of communication obtained by the CCC even though there is overlap in the constructs being measured.  It seems as though the authors are aware of the problems of method variance but have chosen not mention it.   I believe this is too big an issue to be avoided and the authors need to address it and attempt to manage it in their analysis.

Results

In Table 1 a set of t-tests were run comparing the distribution of scores against a reference value based on norms for the test.  Different reference points seem to have been employed for the different measures.  For vocabulary it appears to be the mean (100) whereas others were either 1 or 2 SDs below the mean.   Also, given the large number of tests, presenting a p level seems uninformative.  Either the inferential tests need to be adjusted for the high level of multiple test (given the high correlations it would be best to use an empirical method) or just present the data as description and provide an effect size.

Table 4 presents the results of exploratory factor analysis.  The way the table is organized with two columns running across all measures, it appears that this was a single FA; however in the text, it sounds like separate FAs were run on each set of measures. Thus, what is Factor 1 for the Conners is not the same as for the CCC-2 or the Language measures. At this point, it appears that the authors suspect that is an issue with measurement effects and thus the parent report on the CCC is kept separate from the direct test measures.  This needs to be clearly stated and rationalized.  Also, probably the FAs should each be in a separate table.   In the text, it is stated that 2 factor solutions were chosen.  On what grounds were these chosen?  Was this done via inspection of the scree plot or was a cut off for eigen values used?

The paragraph beginning on line 206, is very hard to follow.  The paragraph begins by describing relationships between the language factors (1: phonological awareness\language and 2: literacy).  It seems to be stating that the two factors were strongly related, but no correlation was given.  Furthermore, these are factor scores and it even though the rotation allowed for correlation it wouldn’t seem likely that they would be highly correlated.  Interestingly, these were apparently not significantly correlated, but no correlation value was provided.  In fact, shouldn’t there be two correlations for each of the language factors (“hot” and “cold” with language and literacy)?  The focus in the paragraph now switches to the measures of communication from the CCC.  In lines 264 to 266 we are told that both the CCC language dimensions were associated with both “hot” and “cold”, but no correlation coefficients were provided. We are then told that for both of the behavior dimensions the pragmatic behaviors were more strongly related than the structural. 

Discussion

The authors note that this study revealed that the communication behaviors on the CCC were associated with both the hot and cold behavioral domain, but that this was not true for language.  The study certainly seems to show that the CCC and the standardized testing of language are different, but the question then is why.  These measures all claim to be measuring a common construct of language.  This would be particularly true with the scales measuring language structure. The obvious explanation for this is that there is a method difference.  Thus, the CCC is very likely more associated with the Conner’s measures because of a common information source – the parent.   This also could explain the paradox of no association between attention and reading.   This issue along with the problems of restriction of range that is mentioned on line 349 m

Author Response

Thank you for your valuable comments on our submission. They have motivated a complete re-think of the paper and re-analysis of the data. We have made three major changes to the manuscript, as a result of comments from both reviewers:

i) In response to concerns raised by both reviewers about how we defined language in our original submission, we have developed the theory in the introduction to make a clear distinction between structural and pragmatic aspects of language function. In doing this we have reframed the manuscript and made it explicit that structural language skills are assessed through parent-ratings of structural communication skills and standardised measures of literacy, while pragmatics are assessed through parent-rated abilities.

ii) Reviewer 1 made an important point that the subgroup comparison did not add to the main message of the paper. The subgroup analyses have been removed to streamline the manuscript.

iii) Both reviewers wanted us to justify our motivation for using exploratory factor analyses and to provide further information about how the different factor solutions were selected for each set of measures. In our previous analysis of the behaviour measures we forced a two-factor solution to separate the two symptom dimensions of ADHD (hyperactivity and inattention). These two factors were very highly correlated (r = .676). For the direct measures of language we forced a two-factor solution to tease apart two distinct features of language attainment (phonological abilities and literacy), and for the parent-ratings of communication we allowed factors to emerge from the data as they were consistent with a distinction between pragmatic and structural communication skills. In the revised analysis we have used an entirely bottom-up approach to allow factor solutions to emerge from the data. This has produced a much simplified pattern of results. A single dimension of behaviour problems was identified. This was strongly associated with a language factor linked with pragmatics, but less so with a structural communication factor and not at all with a factor capturing literacy abilities. We hope the resulting message of the paper is clear: behavioural problems are more strongly related to pragmatic language difficulties than to structural language difficulties.

Due to the substantial revisions we have not included tracked changes on the manuscript. Detailed responses are provided below.

1. The measures used in this study are common in the ADHD literature.  The measures reflect a mixture of direct testing for the language measures and parent report for behavior and communication.  It is very likely that there is a sizable amount of covariance across the parent measures (common method variance) that reflect the parent’s response set and this is not present in the direct measures of language.  Throughout the paper, the authors keep the directly tested language measures separate from the measures of communication obtained by the CCC even though there is overlap in the constructs being measured.  It seems as though the authors are aware of the problems of method variance but have chosen not mention it.   I believe this is too big an issue to be avoided and the authors need to address it and attempt to manage it in their analysis.

                This point has motivated a re-think of the constructs of language being measured in this study. In line with other research into developmental disorders of language, two language constructs (structural and pragmatic) are now identified in the introduction. Conceptually the parent-ratings of structural communication and direct measures of literacy are now both considered measures of the children’s use of language structure, while the parent-ratings of pragmatics are considered as measures of a separate language construct.

To investigate whether the patterns of association between behaviour and communication reflect common method variance, the relationship between both of these factors and the direct measures of language ability are now reported (page 10, line 279-285). These analyses reveal an association between the direct assessments and parent-ratings of language ability, indicating that they are measuring common variance in language function despite one set of assessments relying on subjective reports. There were also important differences in the strengths of the associations between the parent-reports and the direct assessments. First, a stronger relationship was found between parent-rated structural communication and direct literacy measures than between pragmatic communication and literacy. These parent-rated communication scores similarly showed different strengths of association with parents’ behaviour ratings. These two points suggest there is meaningful variation in parent-ratings. Finally, parent ratings of learning were specifically linked with direct measures of literacy, despite there being no other relationships between ratings on other scales of the same behaviour checklist and literacy. Taken together, these features of the data speak against common method variance explaining the links between behaviour and communication. These issues are discussed on page 12, lines 333-347.

Results

2. In Table 1 a set of t-tests were run comparing the distribution of scores against a reference value based on norms for the test.  Different reference points seem to have been employed for the different measures.  For vocabulary it appears to be the mean (100) whereas others were either 1 or 2 SDs below the mean.   Also, given the large number of tests, presenting a p level seems uninformative.  Either the inferential tests need to be adjusted for the high level of multiple test (given the high correlations it would be best to use an empirical method) or just present the data as description and provide an effect size.

                Additional information has been provided to better characterise the sample against age-norms. The sample mean, effect size of the difference between the sample mean and the age-normative mean, and the percentage of children in the deficit range are provided for each measure in Table 2. Across all measures children were classified as being in the deficit range if their scores were 1SD below the age-normed mean. Family-wise corrections for multiple comparisons were applied to all the analyses (see page 8, first paragraph of Analysis Plan, for details).

3. Table 4 presents the results of exploratory factor analysis.  The way the table is organized with two columns running across all measures, it appears that this was a single FA; however in the text, it sounds like separate FAs were run on each set of measures. Thus, what is Factor 1 for the Conners is not the same as for the CCC-2 or the Language measures. At this point, it appears that the authors suspect that is an issue with measurement effects and thus the parent report on the CCC is kept separate from the direct test measures.  This needs to be clearly stated and rationalized.  Also, probably the FAs should each be in a separate table.   In the text, it is stated that 2 factor solutions were chosen.  On what grounds were these chosen?  Was this done via inspection of the scree plot or was a cut off for eigen values used?

                As is now stated explicitly in the Analysis Plan, separate exploratory factor analyses (EFAs) were conducted for each set of measures (behaviour, communication and direct assessments of language). The resulting factor structures are now reported in separate tables (Tables 6, 7, and 8). The behaviour measures were entered into their own EFA. The language measures were entered into two separate EFAs based on the measurement type (parent-ratings or standardised assessments). It was not considered appropriate to enter the different measurement tools into a single EFA as the shared variance within sets of measures would likely dominate the factor solution and overshadow distinctions between the theoretically distinct constructs of language structure and pragmatics. Furthermore, the two sets of measures assessing language structure (direct literacy assessments and parent-rated structural communication skills) measure subtly different abilities; structural language in conversation and the use of structural language for scholastic literacy skills.

To verify the factor structure of the language variables an EFA has been conducted that includes all measures of language function (not reported in the manuscript). A two-factor solution emerged. The pragmatic communication measures of nonverbal communication, inappropriate initiation, social language, use of context, stereotyped language, and interests loaded on Factor 1. There were smaller loadings for three of the structural communication measures (coherence, semantics, speech) on this factor. The direct measures of language (reading, spelling, and vocabulary) and two of the structural communication measures (syntax and speech) loaded on Factor 2. The weightings of the factor loadings are broadly consistent with two language constructs, one related to pragmatic language skills and the other to language structure. However, because three of the structural communication measures also loaded on Factor 1, it is hard to distinguish whether it is tapping general communication skills or pragmatic abilities. To avoid this problem and maximise differences in the language constructs being measured we have retained the outcomes of EFAs that were conducted separately for the communication and direct measures of language in the manuscript. 

In the original analyses a two-factor solution was forced for the behaviour measures to tease apart the symptom dimensions of inattention and hyperactivity. However, in line with the bottom-up approach of the revised manuscript, factor structures emerging from the data were chosen based on both an eigenvalue cut-off of 1 and the scree plots in the new analyses, and no factor solutions were forced. This is stated explicitly in the Analysis Plan section on page 8.

4. The paragraph beginning on line 206, is very hard to follow.  The paragraph begins by describing relationships between the language factors (1: phonological awareness\language and 2: literacy).  It seems to be stating that the two factors were strongly related, but no correlation was given.  Furthermore, these are factor scores and it even though the rotation allowed for correlation it wouldn’t seem likely that they would be highly correlated.  Interestingly, these were apparently not significantly correlated, but no correlation value was provided.  In fact, shouldn’t there be two correlations for each of the language factors (“hot” and “cold” with language and literacy)?  The focus in the paragraph now switches to the measures of communication from the CCC.  In lines 264 to 266 we are told that both the CCC language dimensions were associated with both “hot” and “cold”, but no correlation coefficients were provided. We are then told that for both of the behavior dimensions the pragmatic behaviors were more strongly related than the structural.

                This section has now been fully re-drafted to report the correlations between each of the factors. These are reported on page 10, lines 272-285.

Discussion

5. The authors note that this study revealed that the communication behaviors on the CCC were associated with both the hot and cold behavioral domain, but that this was not true for language.  The study certainly seems to show that the CCC and the standardized testing of language are different, but the question then is why.  These measures all claim to be measuring a common construct of language.  This would be particularly true with the scales measuring language structure. The obvious explanation for this is that there is a method difference.  Thus, the CCC is very likely more associated with the Conner’s measures because of a common information source – the parent.   This also could explain the paradox of no association between attention and reading.   This issue along with the problems of restriction of range that is mentioned on line 349 m

                The manuscript has been substantially revised and now reveals a clearer message: the CCC-2 measures of pragmatic language are associated with behaviour, but the measures of structural language that include the CCC-2 measures of structural communication and the direct measures of literacy are less strongly related to behaviour. In the revised analyses, the CCC-2 and the standardised language measures are not as different as they appeared in the original submission. For example, the standardised language tests and parent-ratings of communicative language are correlated (rs > .300, ps < .001), and there is a significantly stronger relationship between the standardised tests that tap into knowledge of language structure and the CCC-2 ratings of structural than pragmatic language skills. This latter point supports the revised theoretical framework presented in the introduction that distinguishes between structural and pragmatic language abilities.

                As discussed in response to point 1 above and in the Discussion (page 12, lines 336-350), there were meaningful relationships between the parent-ratings and behaviour that suggest the results are not a consequence of common method variance. In particular, the parent and direct assessments of structural language skills were highly related. There were also differences in the strengths of the associations between parent-rated abilities (e.g. pragmatic communication skills were more strongly associated than structural communication skills with behaviour). Similar differences were observed in the relationships between parent-rated behaviour and the standardised measures of language (e.g. parent ratings of learning were specifically linked with direct measures of literacy, despite there being no other relationships between ratings on other scales of the same behaviour checklist and literacy). These findings suggest there is meaningful variation in parent-ratings. 

The lack of an association between parent-rated inattention and literacy, which has been reported previously for typically developing or specific diagnostic groups, was surprising. It is unlikely to be the result of different sources of variance (parent vs direct assessment) as a similar pattern was detected in the parent ratings of language. In line with the direct assessments, there was a weak relationship between parent-ratings of structural language skills and inattention. This is the first large-scale investigation into the relationship between inattention and the use of structural language skills in a highly atypical and transdiagnostic group, which may explain why the associations are different to those reported previously. It is suggested that the absence of this association in the current sample might reflect the fact that behaviour and literacy problems arise through different sources in sub-clinical populations, or that weak literacy skills are underpinned by additional cognitive impairments that do not cause behavioural difficulties (e.g. phonological processing problems).  This point is discussed on pages 11-12, lines 318-332. 

Round 2

Reviewer 2 Report

The revision was very responsive to the prior review.